# Hydroxyurea modulates thiol–disulfide homeostasis in the yeast endoplasmic reticulum

Yuki Takano[1], Yuki Ishiwata-Kimata[2], Ryo Ushioda[3,4], Yukio Kimata[2], Kunio Nakatsukasa[1]

**Hydroxyurea (HU) has been extensively used in laboratory settings to induce S-phase arrest and checkpoint activation. Furthermore, it has a history of clinical use as a cost-effective chemotherapeutic agent. Nevertheless, there is still uncertainty regarding its precise pharmacology, side effects, and toxicity, particularly in terms of its impact on organelle homeostasis. Here, we demonstrate that in budding yeast, HU specifically inhibits the endoplasmic reticulum–associated degradation (ERAD) of luminal misfolded proteins (ERAD-L pathway), an effect that is independent of S-phase arrest. In contrast, HU did not affect the degradation of misfolded ER membrane proteins or the degradation of cytosolic proteins. The selective inhibition of ERAD-L by HU is likely attributable to the formation of disulfide bonds in cysteine residues in luminal substrates, which must be reduced before their retrotranslocation to the cytosol. We further demonstrate that HU plays a role in alleviating reductive stress phenotypes observed in cells lacking Ero1, which is essential for oxidative protein folding in the ER. We propose that HU functions as a modulator of thiol–disulfide homeostasis in the ER lumen.**

## Introduction

Newly synthesized secretory and membrane proteins are translocated into the endoplasmic reticulum (ER), where they are directed into the secretory pathway. The ER harbors a high concentration of molecular chaperones that maintain polypeptide solubility, enzymes that catalyze post-translational modifications, and factors that facilitate oxidative folding by promoting the formation of disulfide bonds between cysteine residues in newly synthesized polypeptides. In budding yeast *Saccharomyces cerevisiae*, the key components of the oxidative protein folding pathway in the ER are protein disulfide isomerase (Pdi1) and ER oxidase Ero1 (Frand & Kaiser, 1998, 1999; Pollard et al, 1998; Cuozzo & Kaiser, 1999; Tu et al, 2000; Robinson & Bulleid, 2020; Wang & Wang, 2023). This oxidative folding pathway consists of a linear electron transfer cascade, where electrons from the thiol (SH) groups of reduced proteins are transferred via Pdi1 to Ero1 through a series of thiol oxidoreductions. The electrons ultimately flow to a bound flavin adenine dinucleotide group within Ero1 and are then transferred to molecular oxygen ($O_2$) (Robinson & Bulleid, 2020; Wang & Wang, 2023). The net outcome of this process is the formation of disulfide-bonded proteins and the production of hydrogen peroxide.

In the event of an accumulation of proteins that cannot attain their native conformation in the ER, a cellular adaptive response termed the unfolded protein response (UPR) is activated to restore ER homeostasis (Mori, 2009; Karagoz et al, 2019). In *S. cerevisiae*, the UPR is initiated by the ER stress sensor Ire1, which undergoes phosphorylation upon oligomerization, thereby activating its endoribonuclease activity. This leads to the splicing and activation of the transcription factor Hac1, which subsequently up-regulates the expression of genes involved in restoring ER homeostasis (Travers et al, 2000; Mori, 2009; Karagoz et al, 2019). However, terminally misfolded ER proteins are specifically recognized and retrotranslocated to the cytosol where they are ubiquitinated and targeted to the proteasomal degradation, a process known as ER-associated degradation (ERAD) (Ruggiano et al, 2014; Wu & Rapoport, 2018; Mehrtash & Hochstrasser, 2019; Kumari & Brodsky, 2021; Ninagawa et al, 2021; Christianson et al, 2023). ERAD substrates with luminal lesions are directed to the Hrd1 ubiquitin ligase complex (ERAD-L pathway) (Huyer et al, 2004; Vashist & Ng, 2004; Carvalho et al, 2006; Stein et al, 2014; Baldridge & Rapoport, 2016), whereas membrane proteins with lesions within their membrane-spanning regions are also targeted to the same ligase (ERAD-M pathway) (Carvalho et al, 2006; Sato et al, 2009; Neal et al, 2018). In contrast, membrane proteins with misfolded lesions exposed to the cytosol are predominantly recognized by Doa10, another integral membrane ubiquitin ligase in the ER membrane (Huyer et al, 2004; Vashist & Ng, 2004; Carvalho et al, 2006; Ravid et al, 2006). Ubiquitinated substrates are then segregated from the ER membrane and delivered to the proteasome for degradation (Jentsch & Rumpf, 2007; Baldridge & Rapoport, 2016).

---

[1]Graduate School of Science, Nagoya City University, Nagoya, Japan   [2]Division of Biological Science, Graduate School of Science and Technology, Nara Institute of Science and Technology, Nara, Japan   [3]Laboratory of Molecular and Cellular Biology, Department of Frontier Life Sciences, Faculty of Life Sciences, Kyoto Sangyo University, Kyoto, Japan   [4]Institute for Protein Dynamics, Kyoto Sangyo University, Kyoto, Japan

Correspondence: nakatsukasa@nsc.nagoya-cu.ac.jp

Hydroxyurea (HU), also known as hydroxycarbamide, is a cost-effective chemotherapeutic in clinical use and is widely employed to induce S-phase arrest and checkpoint activation in the laboratory. The primary mechanism of HU's action has been postulated to be its ability to inhibit DNA replication by inactivating ribonucleotide reductase (RNR), thereby depleting the pools of deoxyribonucleotide triphosphates (dNTPs) (Young & Hodas, 1964; Krakoff et al, 1968). However, HU has also been implicated in the generation of reactive oxygen species (ROS), likely through radical chain reactions initiated by its hydroxylamine group (Sakano et al, 2001; Hammond et al, 2003; Juul et al, 2010; Huang et al, 2016; Singh & Xu, 2016; Kapor et al, 2021). Notably, ROS generated by HU reportedly inhibits the polymerase activity of replicative polymerases, possibly by oxidizing their iron–sulfur (Fe-S) clusters, leading to polymerase complex dissociation and a subsequent loss of DNA substrate binding, thus triggering S-phase arrest (Shaw et al, 2024). This result may help explain the previous observation that the replication arrest is induced in HU-treated cells even when basal levels of dNTPs are maintained (Koc et al, 2004). In this way, the role of ROS and their downstream targets in the cytotoxicity of HU has been suggested; however, the understanding of HU's mechanisms of action, pharmacology, side effects, and toxicity, particularly for the impact on organelle homeostasis, remains incomplete.

In the course of investigating the potential regulation of ERAD during the cell cycle in yeast, we unexpectedly discovered that HU induces the formation of disulfide bonds in ER luminal proteins and specifically inhibits ERAD-L, an effect that is independent of S-phase arrest. HU did not affect ERAD-M, ERAD-C, or the degradation of cytosolic proteins. Moreover, we found that thiol oxidation induced by HU can alleviate the reductive stress phenotypes associated with an Ero1 mutant, including ER-to-Golgi transport defects and cellular growth defects. HU has been a well-established treatment for a range of diseases. This study suggests that HU is a drug that acts to modulate thiol–disulfide homeostasis in the ER, and may provide a novel perspective on understanding HU's pharmacology and mechanisms of action on various diseases.

# Result

### HU specifically inhibits ERAD-L independently of S-phase arrest

We initially sought to examine whether endoplasmic reticulum–associated degradation (ERAD) is physiologically regulated during the cell cycle in budding yeast. To monitor ERAD in cells synchronized at the G1, S, or G2/M phase, we treated cells with α-factor, HU, or nocodazole (NC), respectively, and performed cycloheximide chase analysis of model ERAD substrates (Figs 1 and S1, and Table S1). The cell cycle synchronization was confirmed by the levels of Sic1 and Clb2 (Fig S2A and B), which peak at G1 and G2/M phases, respectively (Visintin et al, 1998). Model ERAD-L substrates including CPY* (Finger et al, 1993), KHN (Vashist & Ng, 2004), and KWW (Vashist & Ng, 2004) were degraded in cells treated with α-factor or NC to a similar extent as in control cells. However, their degradation was considerably inhibited in cells treated with HU (Figs 1A and S1A and B). The inhibition of ERAD-L by HU was dose-dependent, with an

inhibitory effect observed at the lowest concentration of ~1 mg/ml (~13 mM) (Fig S3). In contrast, the turnover of model ERAD-M substrates including Pdr5* and 6myc-Hmg2 (Hampton et al, 1996; Egner et al, 1998; Plemper et al, 1998) was unaffected by all of these drugs (Figs 1B and S1C). The turnover of model ERAD-C substrates including Ste6* (Loayza et al, 1998) and Pca1 (Adle et al, 2009) was also unaffected by all of these drugs (Figs 1C and S1D). Finally, cytosolic soluble substrates including NΔCit1 (Nishio et al, 2023), Cit2-GFP-SKL (Nakatsukasa et al, 2015), and Spo12 (Nakatsukasa et al, 2018) were also normally degraded in cells treated with HU (Fig S4A–C). These results demonstrate that HU specifically inhibits the degradation of ERAD-L substrates, but not of ERAD-M, ERAD-C, and cytosolic substrates.

To investigate whether S-phase synchronization is a prerequisite for HU-mediated inhibition of ERAD-L, we analyzed CPY* degradation under several conditions. First, we found that CPY* degradation was strongly inhibited even in cells treated with HU only for 15 min (Fig 2A, lanes 17–20). This is much shorter than ~2.5 h, which is generally needed to fully synchronize the cell cycle of budding yeast. Second, we analyzed the level of CPY* over time during the cell cycle. Cells were synchronized at the G1 phase by treatment with α-factor, and then, the cell cycle was restarted by removing α-factor. Cell cycle progression was confirmed by the levels of Sic1 and Clb2 (Fig 2B, αSic1 and αClb2). The level of CPY* was unchanged during the cell cycle (Fig 2B, αHA). Third, under our experimental conditions, cells entered the S phase at 30 min after the cell cycle was restarted from the G1 phase (Fig 2B, lane 4, indicated by an arrow; Fig 2C, upper panel, lane 6). In these cells, the initial rate of CPY* degradation was faster than that in cells treated with HU (Fig 2C, lower panel, αHA, lanes 9–12). Fourth, the DNA-damaging alkylating agent methyl methanesulfonate (MMS), which also induces genotoxic stress and activates the S-phase checkpoint, did not stabilize CPY* (Fig 2D). It should be noted that when cells were treated with MMS or HU under the conditions of Fig 2D, we did indeed observe phosphorylation of Rad53 (Fig 2E), indicating that MMS treatment in Fig 2D actually activated the S-phase checkpoint. However, CPY* degradation was unaffected in cells treated with MMS, whereas it was considerably inhibited in HU-treated cells (Fig 2D). Together, these results suggest that HU inhibits ERAD-L independently of S-phase arrest.

### The ERAD machinery is intact in HU-treated cells

To investigate the mechanism by which HU inhibits ERAD-L, we first asked if HU treatment alters the integrity of the Hrd1 complex (Fig S5A). The functional tagged form of Hrd1-3FLAG (Nakatsukasa et al, 2013) was immunoprecipitated from digitonin-solubilized membranes prepared from cells treated with HU. The components of the Hrd1 complex including Usa1, Hrd3, and Der1 and Yos9 (Carvalho et al, 2006; Denic et al, 2006; Gauss et al, 2006; Christianson et al, 2011) were all co-precipitated to the same extent from HU-treated and untreated cells (Fig S5B). Der1 was expressed slightly more in HU-treated cells than in nontreated cells (Fig S5B, compare lanes 2 and 3, αDer1), but previous studies suggested that the overexpression of Der1 does not significantly interfere with the degradation of luminal substrates (Horn et al, 2009; Carroll & Hampton, 2010; Carvalho et al, 2010; Mehnert et al, 2014). Therefore, induction

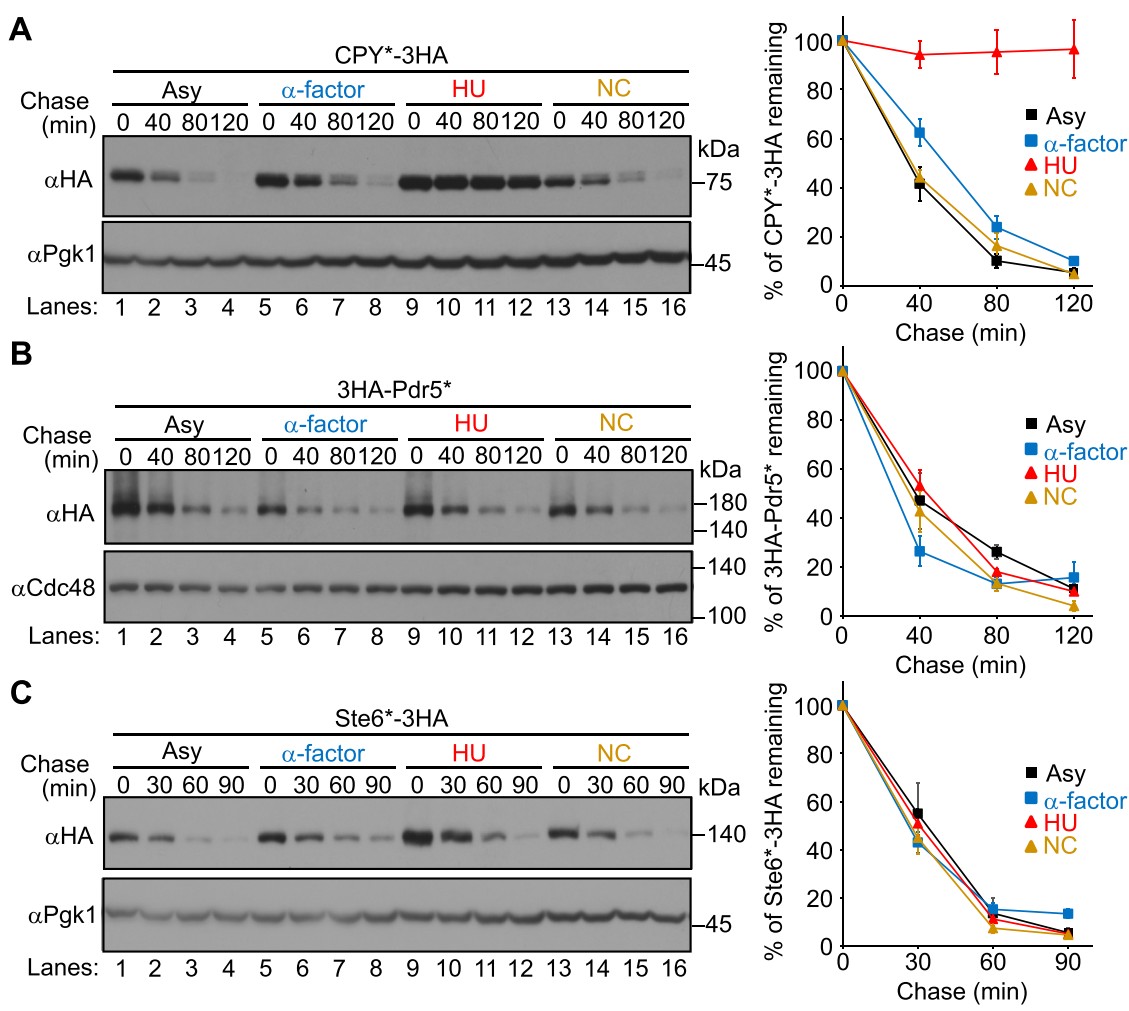

**Figure 1. Cycloheximide chase analysis of ERAD substrates in cells synchronized at the G1, S, or G2/M phase.**
**(A)** WT cells expressing CPY*-3HA were grown at 30°C in SD+DO+5xAde and shifted to YPD+5xAde before the cell cycle was synchronized at the G1, S, or G2/M phase by incubation with α-factor, HU, or NC, respectively, for 2.5 h. Cycloheximide (200 μg/ml) was added, and cells were collected at the indicated time points. Total cell lysates were subjected to Western blotting with an anti-HA antibody. **(A, B, C)** Cycloheximide chase analyses of 3HA-Pdr5* and Ste6*-3HA in cells arrested at the G1, S, or G2/M phase were performed as in (A). Expression of Ste6*-3HA was induced under the control of the *GAL1* promoter in the medium containing 2% galactose as a sole carbon source. Pgk1 or Cdc48 served as loading controls. Signals were normalized to those for loading controls. Quantification of the results is shown. The data represent the mean ± SE of three independent experiments.
Source data are available for this figure.

of Der1, if any, did not explain HU-mediated inhibition of ERAD-L. These results suggest that HU does not affect the integrity of the Hrd1 complex.

We next analyzed substrate recognition in cells treated with HU. Yos9 and Hrd3 recognize ERAD-L substrates independently of each other before substrates are retrotranslocated through Hrd1/Der1 channel (Denic et al, 2006; Wu et al, 2020). When CPY* was immunoprecipitated from the digitonin-solubilized membrane fraction prepared from HU-treated *hrd3Δ* cells, Yos9 was co-immunoprecipitated to the same extent as from nontreated *hrd3Δ* cells (Fig S5C). Similarly, Hrd3 was co-immunoprecipitated with CPY* to the same extent from HU-treated and nontreated *yos9Δ* cells (Fig S5D). In addition, the recognition of CPY* by Kar2 (the ER luminal Hsp70, BiP) was also unaffected (Fig S5E), and CPY* remained soluble in cells treated with HU (Fig S5F). These results suggest that HU does not affect

Yos9/Hrd3/BiP-mediated recognition and the solubilities of ERAD-L substrates.

We noticed that in cells treated with HU, the ER-to-Golgi transport was partially, albeit not completely, compromised. For example, it has been reported that CPY* acquires α1,6 mannose on its *N*-glycans in the *cis*-Golgi compartment upon inhibition of ERAD. This is because this substrate is recycled between the ER and the *cis*-Golgi upon ERAD inhibition (Caldwell et al, 2001). This phenomenon can be observed by noting the slight shift in the position of the CPY* band derived from *hrd1Δ* cells (Fig S6A, lanes 9–12) on an SDS–PAGE gel. However, the aforementioned shift was barely discernible for CPY* in HU-treated cells, despite the inhibition of its degradation (Fig S6A, lanes 5–8). A more evident illustration is KHN; its *O*-linked mannose chain undergoes elongation during its recycling between the ER and the *cis*-Golgi apparatus when ERAD is

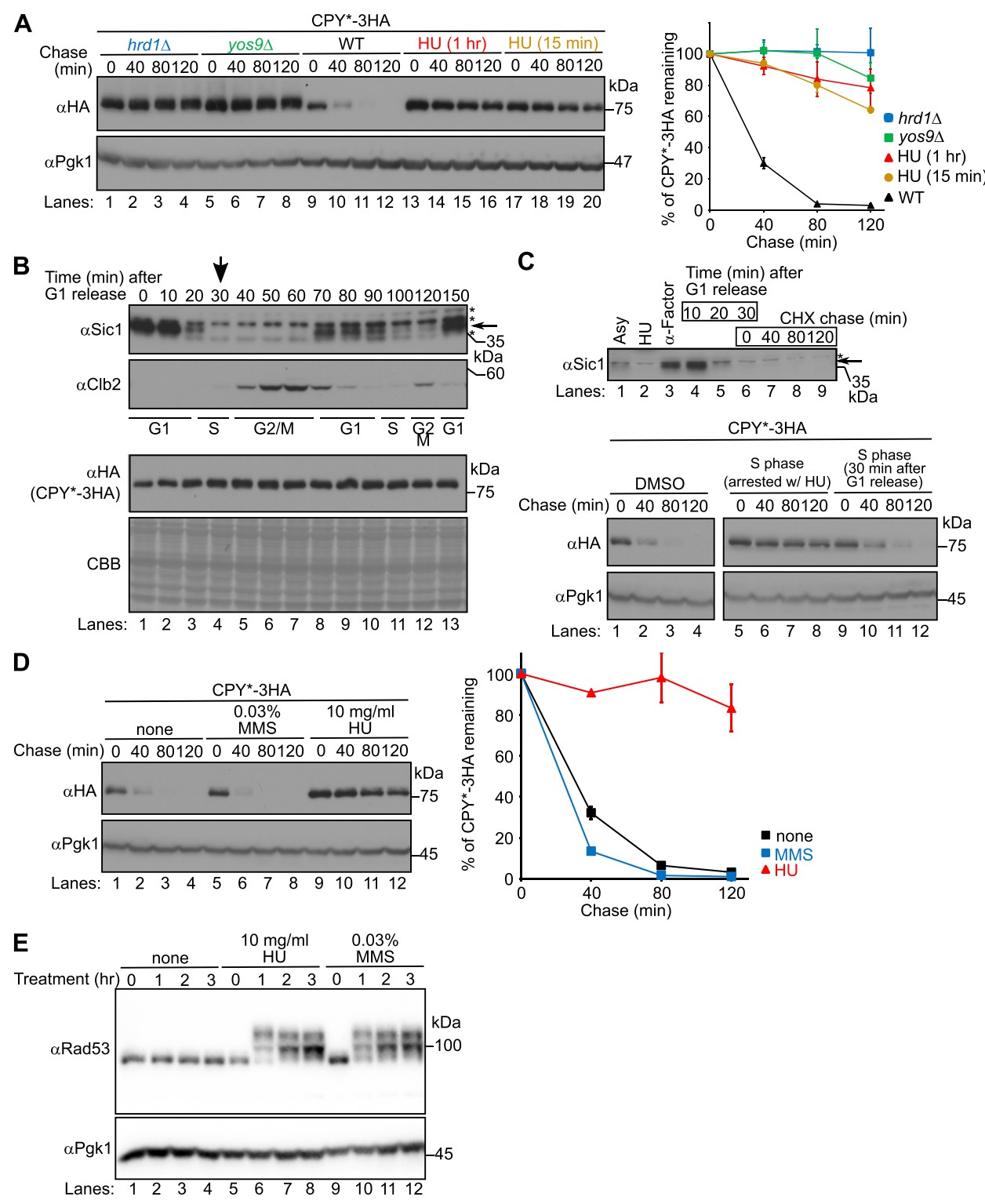

**Figure 2. HU inhibits ERAD-L independently of S-phase arrest.**

**(A)** Cycloheximide chase analysis of CPY*-3HA was performed as in Fig 1A. Where indicated, cells were treated with 10 mg/ml HU for 1 h (lanes 13–16) or 15 min (lanes 17–20) before the addition of cycloheximide. CPY*-3HA signals were normalized to Pgk1 signals. **(B)** Cells expressing CPY*-3HA were grown at 30°C until OD_600 reached ~0.35 before being synchronized at the G1 phase in YPD+5xAde with α-factor. After 2.5 h, α-factor was removed and the cell cycle was restarted in fresh medium (0 min). At the indicated time points, an equal number of cells were collected by centrifugation and subjected to Western blotting with an anti-Sic1, anti-Clb2, or anti-HA (CPY*) antibody. Coomassie brilliant blue (CBB) staining of the membrane served as a loading control. **(C)** Cells that entered the S phase (indicated by the arrow) were used for the cycloheximide chase analysis in (C). **(B, C)** Cells expressing CPY*-3HA were grown to a log phase (upper panel, lane 1) and arrested at the G1 phase (lane 3), and the cell cycle was restarted as in (B). After 30 min (lane 6), when cells entered the S phase, cycloheximide chase analysis of CPY* was performed as in Fig 1A (lower panel, lanes

inhibited, and the bands for KHN are reportedly shifted upward during the chase period (Vashist et al, 2001) (Fig S6B, lanes 9–12). However, although the elongation of the O-linked mannose chain was observed in HU-treated cells, it was not as pronounced as in hrd1Δ cells (Fig S6B, lanes 5–6). The inefficient O-linked chain elongation observed in HU-treated cells was rescued by the administration of NAC (Fig S6C, lanes 13–16). Nevertheless, the extent of HU-induced inhibition of the ER-to-Golgi transport was markedly less pronounced than that observed in sec12-4 mutant cells (Fig S6D, lanes 5–8), in which ERAD-L is almost entirely suppressed (Vashist et al, 2001). Consequently, the retardation of ER-to-Golgi transport does not appear to be a primary factor contributing to the selective inhibition of ERAD-L by HU.

### HU leads to the formation of disulfide bonds in the ER luminal proteins

It has been reported that treatment of cells with HU may result in the accumulation of ROS (Sakano et al, 2001; Hammond et al, 2003; Juul et al, 2010; Singh & Xu, 2016; Kapor et al, 2021). In line with this idea, treatment of cells with N-acetylcysteine (NAC), an anti-oxidative reagent, rescued the degradation defects of CPY* and KHN in cells treated with HU (Fig 3A, lanes 13–16, and Fig S7, lanes 13–16). This suggests that HU-mediated oxidation is a potential cause of ERAD-L inhibition. A recent study has also indicated that in budding yeast, HU generates $H_2O_2$, which in turn likely inactivates iron–sulfur cluster–containing proteins such as DNA polymerase, and arrests the cell cycle at the S phase (Shaw et al, 2024). However, treatment of cells with different concentrations of $H_2O_2$ inhibited not only ERAD-L but also ERAD-M, ERAD-C, and cytosolic degradation pathways (Fig S8A–F), indicating that $H_2O_2$ generated by HU administration and the externally added $H_2O_2$ lead to disparate effects on ERAD. In contrast, treatment of cells with diamide, which oxidizes thiols and facilitates the formation of disulfide bridges between cysteine residues, specifically inhibited ERAD-L but not ERAD-M, ERAD-C, or cytosolic pathways (Fig S9A–F). The selective inhibition of ERAD-L upon perturbation of cysteine redox homeostasis was also corroborated by the observation that ERAD-L was specifically inhibited in cells expressing mutant form of Pdi1 with mutations at one of two thioredoxin-like domains (Holst et al, 1997) (Fig S10A–F).

Based on these observations, we hypothesized that the treatment of cells with HU results in the oxidation of thiols in the ER luminal proteins including ERAD-L substrates, whose disulfide bonds should be reduced before their retrotranslocation to the cytosol (Zhao et al, 2025). We thus assessed the oxidation state of CPY* in cells. To this end, cells were treated with trichloroacetic acid (TCA) and the extracted proteins were subsequently treated with the thiol-conjugating reagent maleimide-PEG5000 (Tsai & Rapoport, 2002; Sakoh-Nakatogawa et al, 2009). In cells that had

not been treated with HU, most of CPY* was in a partially oxidized/reduced form (Fig 3B, lane 9). In contrast, CPY* was resistant to modification by maleimide-PEG5000 in HU-treated cells, indicating that the cysteine residues in CPY* had been almost completely oxidized by HU (Fig 3B, lane 10). When TCA-precipitated proteins prepared from HU-treated cells were preincubated with dithiothreitol (DTT) before being modified with maleimide-PEG5000 (Fig S11A, bottom scheme), thiol-modified forms of CPY* with slower mobility were observed (Fig S11B, compare lanes 11–13 and 14–16). The same outcome was observed when cells were treated with diamide (Fig S11C, compare lanes 9–10 and 11–12). Therefore, it was confirmed that HU-mediated oxidation resulted in the formation of a disulfide bond between cysteine residues in this substrate. The administration of NAC to cells that had been treated with HU resulted in the slower migration of CPY* (Fig 3B, lane 8), suggesting that HU-mediated oxidation was reversed by the NAC treatment. This result is consistent with our observation that treatment of cells with NAC rescued CPY* degradation (Fig 3A, lanes 13–16).

We also analyzed the oxidation state of Pdi1 and a GFP-based redox sensor, FROG/B (Sugiura et al, 2020), through the maleimide-PEG5000 modification assay. FROG/B was fused with the Kar2 (BiP in yeast) signal sequence at its N terminus and with the ER retention signal of HDEL at its C terminus, resulting in the creation of the ER-localized BiPss-FROG/B-HDEL. The results were essentially identical to those obtained with CPY*. Both Pdi1 and BiPss-FROG/B-HDEL were observed to be in a partially oxidized/reduced form in untreated cells (Fig 3C, lane 9; Fig S12A, lane 8), and they were converted to the oxidized form when cells were treated with HU (Fig 3C, lane 10; Fig S12A, lane 9). When protein samples prepared from HU-treated cells were preincubated with DTT before maleimide-PEG5000 modification, the formation of thiol-modified forms of Pdi1 and BiPss-FROG/B-HDEL with slower mobility was observed (Fig S11D, lanes 14–16; Fig S12A, lane 10). The same outcomes were observed when cells were treated with diamide (Figs S11E and S12B). In light of these findings, we propose that HU induces the formation of disulfide bonds in a diverse array of ER luminal proteins.

Given the fact that HU treatment resulted in the formation of disulfide bridges between cysteine residues in CPY* and a subsequent slowing of its degradation, it may be anticipated that the replacement of these cysteine residues with alanine would lead to a robust degradation in cells treated with HU. However, although the mutant (CPY*CysLess) was degraded faster than CPY* in untreated cells, it was weakly stabilized in HU-treated cells, although to a lesser extent than CPY* (Fig S13A). Our initial hypothesis was that HU-mediated oxidation compromised the functional unit formed by Pdi1 and the mannosidase Mnl1/Htm1, a yeast homolog of mammalian EDEM1 (Hosokawa et al, 2001; Jakob et al, 2001; Nakatsukasa et al, 2001; Sakoh-Nakatogawa et al, 2009; Zhao et al, 2025). However, we consider this hypothesis to be implausible. The degradation of CPY*CysLess was indeed dependent on Mnl1

9–12). Cell cycle arrest and progression were monitored by the level of Sic1 (upper panel). Cycloheximide chase analysis of CPY* in cells arrested at the S phase by treatment with HU for 2.5 h was also performed (lower panel, lanes 5–8). **(A, D)** Cycloheximide chase analysis of CPY*-3HA in cells treated with 10 mg/ml HU or 0.03% methyl methanesulfonate for 1 h was performed as in (A). Pgk1 served as a loading control. Signals were normalized to those for loading controls. **(E)** Phosphorylation state of Rad53 was analyzed in cells treated with the indicated concentrations of methyl methanesulfonate or HU. Pgk1 served as a loading control. Quantification of the results represents the mean ± SE of three independent experiments.
Source data are available for this figure.

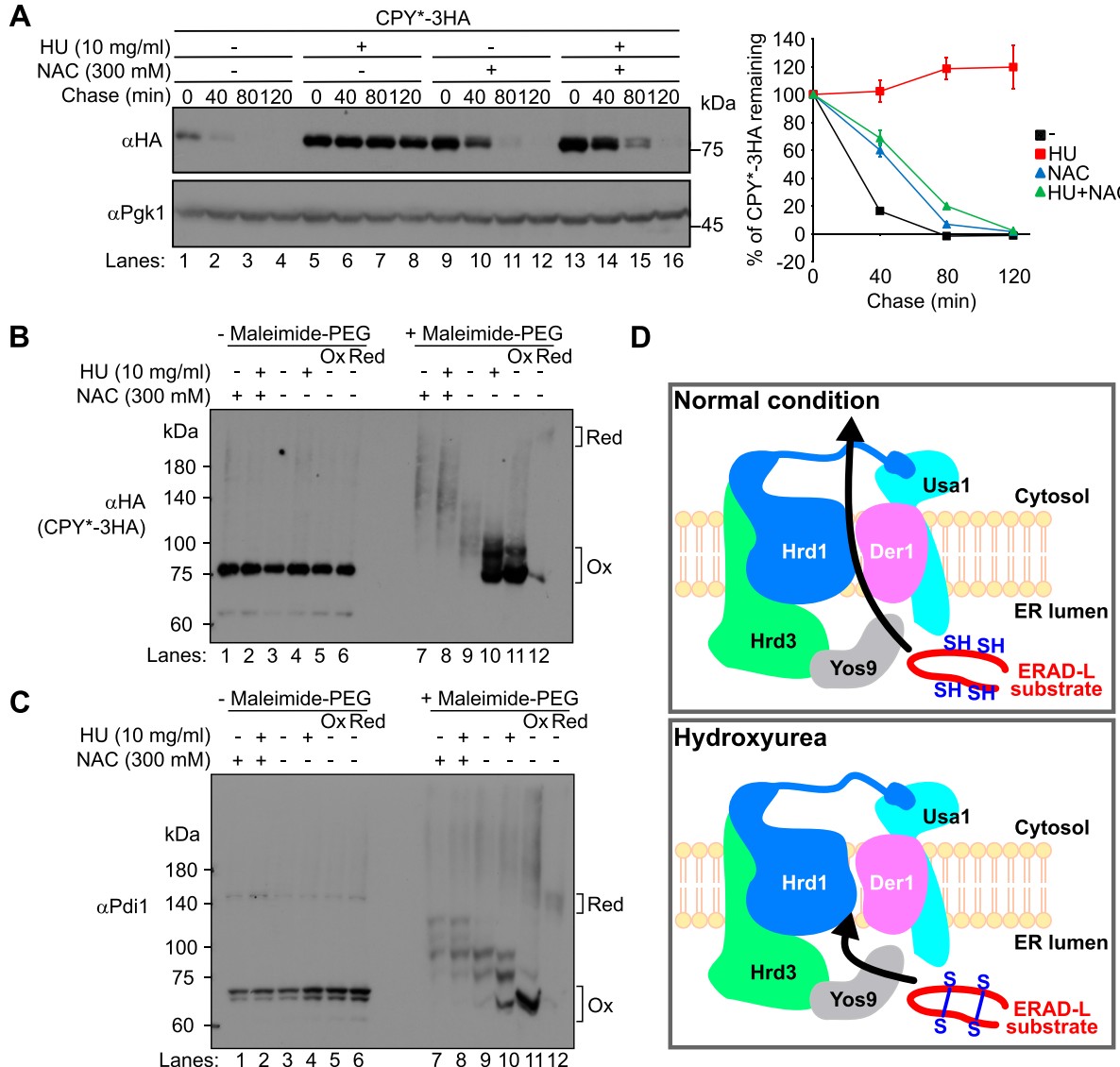

**Figure 3. HU oxidizes cysteine residues and facilitates disulfide bond formation in the ER luminal proteins.**
**(A)** Cycloheximide chase analysis of CPY*-3HA was performed as in Fig 1A. Where indicated, cells were grown to a log phase at 30°C and treated with 10 mg/ml HU for 2 h. Where indicated, 300 mM N-acetylcysteine (NAC) was added to the medium 1 h after the addition of HU. Cycloheximide was added to the medium, and cells were collected at the indicated time points. CPY*-3HA signals were normalized to Pgk1 signals. Quantification of the results is shown. The data represent the mean ± SE of three independent experiments. **(B, C)** Cells were treated with 10 mg/ml HU for 30 min. Where indicated, 300 mM NAC was added to the medium 15 min after the addition of HU. Proteins were precipitated with trichloroacetic acid and modified with maleimide-PEG5000. **(B, C)** The samples were analyzed by SDS–PAGE and immunoblotting with anti-HA (B) or anti-Pdi1 (C) antibodies. As an oxidized and reduced protein control, the trichloroacetic acid–precipitated proteins were pretreated with either 100 mM DTT or 5 mM 5,5′-dithiobis (2-nitrobenzoic acid) before modification reaction with maleimide-PEG5000. **(D)** ERAD-L substrates are reduced before retrotranslocation through Hrd1/Der1 channel to the cytosol. In HU-treated cells, these substrates are oxidized and cannot be delivered to the cytosol for degradation. Source data are available for this figure.

(Pfeiffer et al, 2016; Zhao et al, 2025) (Fig S13B), yet the function of Mnl1 appeared to be intact, even if not complete, because HU had no effect on the turnover of Mnl1-dependent substrate Pdr5* (Jakob et al, 2001) (Fig S13C). Note that Pdr5* degradation was slowed in cells expressing the thioredoxin-like domain mutant form of Pdi1, which likely perturbs the Mnl1 function (Fig S10E). It is rather hypothesized that in cells treated with HU, endogenous, oxidized ERAD substrates are recognized by the ERAD machinery in a manner analogous to CPY* (Fig S5C and D). But they cannot be retrotranslocated through the channel because their cysteine residues form disulfide bonds, which must be reduced before retrotranslocation through the channel formed by Hrd1/Der1 (Mehnert et al, 2014; Wu et al, 2020). Consequently, these substrates accumulate at the luminal side of the channel, which impedes the retrotranslocation of CPY*CysLess (Fig S13A). Consistent with this idea, CPY*CysLess was weakly stabilized in diamide-treated cells to the same extent as in cells treated with HU (Fig S13A). We also note that CPY*CysLess degradation was considerably

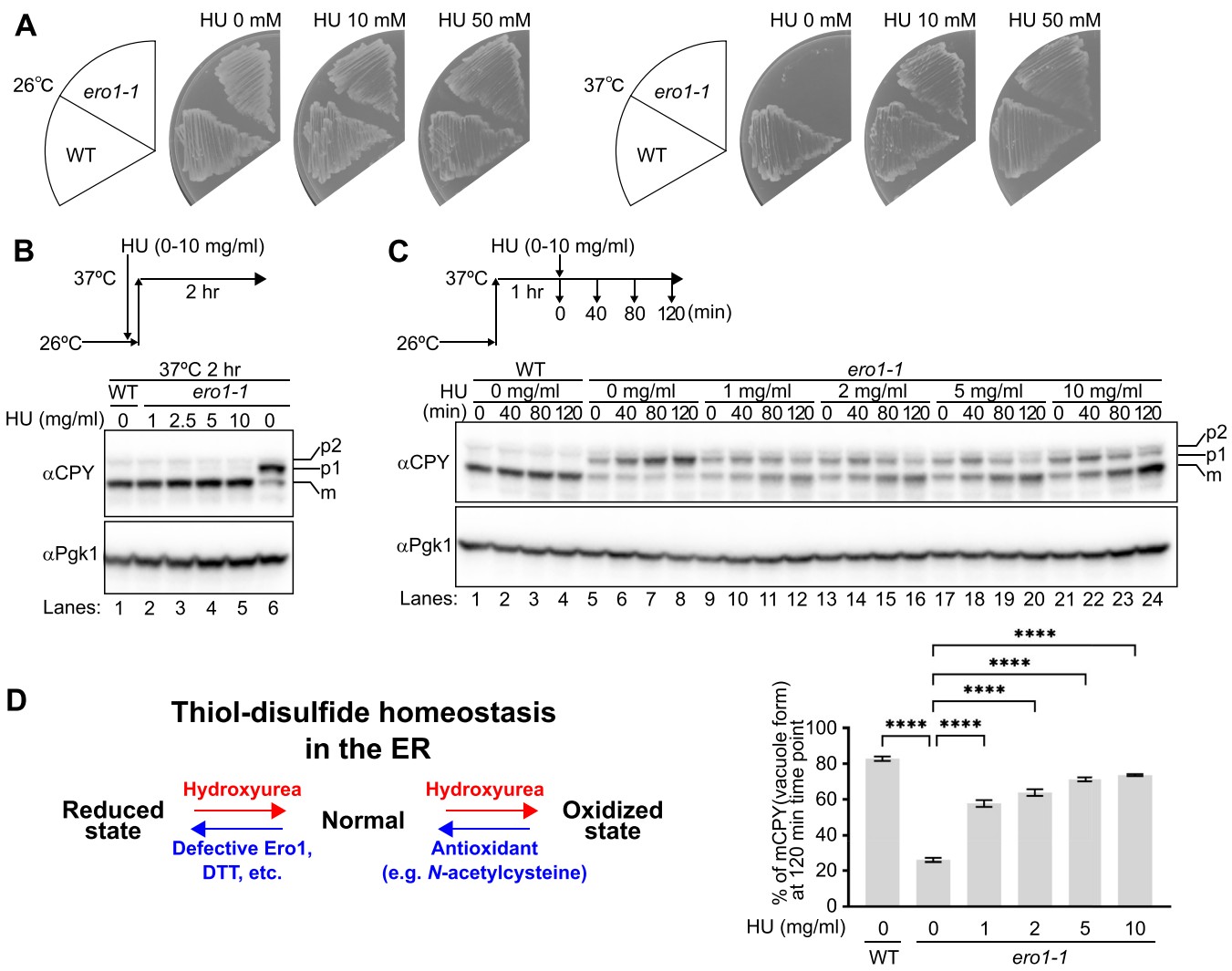

**Figure 4. HU alleviates the ER reductive stress.**
**(A)** *ero1-1* temperature-sensitive strain and its isogenic WT strain were grown on YPD+5xAde plates containing 0, 10 mM (0.76 mg/ml), or 50 mM (3.8 mg/ml) of HU at 26°C or 37°C (nonpermissive temperature of *ero1-1* allele) for 2 d. **(B)** *ero1-1* and WT cells were grown at 26°C and incubated with the indicated concentrations of HU before they were shifted to 37°C for 2 h. The transport of CPY was analyzed by Western blotting with specific antibodies. The ER form (p1), Golgi form (p2), and vacuole form (m) of CPY are indicated. Pgk1 served as a loading control. **(C)** *ero1-1* and WT cells were grown at 26°C and shifted to 37°C for 1 h before they were incubated with the indicated concentrations of HU. **(B)** Cells were collected at the indicated time points, and the transport of CPY was analyzed as in (B). The ratio of mature (vacuole) form to the total amount of CPY was quantified and shown in the graph. Quantified data represent the mean ± SE of three independent experiments. ****$P$ < 0.0001; the $P$-value was calculated by one-way ANOVA with Dunnett's test. **(D)** Schematic representation of the proposed action of HU. HU modulates thiol–disulfide homeostasis in the ER. Source data are available for this figure.

slowed in cells treated with $H_2O_2$ (Fig S13A). This is likely because $H_2O_2$ inhibits the degradation of all pathways tested in this study; not only ERAD-L but also ERAD-M, ERAD-C, and cytosolic proteins. Collectively, we propose that the cysteine oxidation induced by HU administration is a pivotal factor contributing to the selective inhibition of ERAD-L (Fig 3D).

## HU alleviates the ER reductive stress

The effect of HU on the ER luminal proteins was found to be almost similar to that of diamide. It has been demonstrated that diamide is capable of reversing the oxidative protein folding defect resulting from the dysfunction of Ero1, a thiol oxidase within the ER (Frand &

Kaiser, 1998, 1999; Pollard et al, 1998; Cuozzo & Kaiser, 1999; Tu et al, 2000). We thus sought to determine whether HU could also mitigate the oxidative folding defect observed in the *ero1-1* strain, which harbors a G229S mutation that destabilizes the binding of Ero1 to a flavin cofactor at the restrictive temperature, and likely impairs its ability to oxidize PDI (Frand & Kaiser, 1998; Gross et al, 2004; McCulley et al, 2014).

First, HU was able to rescue the temperature-sensitive growth defect observed in *ero1-1* cells (Fig 4A) (McCulley et al, 2014). The minimal concentration was 1 mM (Fig S14A). In the absence of Pdr5, a drug transporter, 1 mM HU, exhibited a slightly more pronounced effect on rescuing the growth defect. This phenomenon can be attributed to the absence of exportation of HU from the cells in the

absence of Pdr5, resulting in its accumulation within the cell. Interestingly, at high concentrations of HU (5–10 mM), the growth levels of the *ero1-1pdr5Δ* cells were found to be lower than that of *ero1-1* cells (Fig S14A). One potential explanation for this phenomenon is that the high concentration of HU caused delayed S-phase progression in *pdr5Δ* cells with greater efficiency compared with *PDR5* cells. However, *ero1-1* and *ero1-1pdr5Δ* cells exhibited comparable growth patterns even in the presence of 5~10 mM HU at a permissive temperature of 26°C, implying that there is not a considerable difference in the delayed S-phase progression between these strains. Regardless of the precise reason for this phenomenon, it is posited that the deletion of a drug transporter such as Pdr5 may exert a more potent effect on the efficacy of HU as an oxidant for the ER luminal proteins.

Second, HU was observed to suppress the UPR in *ero1-1* cells and in cells treated with DTT (Fig S14B–E). Intriguingly, HU itself did not induce the UPR, whereas diamide did (Fig S14F). This was the sole distinction between HU and diamide. Consistent with these observations, at concentrations of diamide as low as 27.8 and 166.7 μM, diamide was observed to suppress the UPR of *ero1-1* cells. Intriguingly, at the higher concentration (1,000 μM), it caused the UPR of *ero1-1* cells (Fig S14G). In case of diamide, a threshold of ~200 μM may exist. At concentrations lower than the threshold, the oxidative capacity of diamide may function to rescue the reductive stress of *ero1-1* cells. However, at higher concentrations, excess diamide may excessively induce oxidation of the ER luminal proteins in *ero1-1* cells, consequently activating the UPR. Nonetheless, HU was observed to rescue the defect in the ER-to-Golgi transport of CPY in *ero1-1* cells at a nonpermissive temperature (Figs 4B and S15). Furthermore, HU was capable of restoring the transport of CPY that had accumulated in the ER of *ero1-1* cells at a nonpermissive temperature (Fig 4C). Based on these findings, we propose that HU is a drug that acts to modulate thiol–disulfide homeostasis in the ER (Fig 4D).

## Discussion

Hydroxyurea (HU) has a long history of use in laboratory settings for the purpose of arresting the cell cycle in the S phase. However, its precise pharmacology, side effects, and toxicity, particularly for the impact on organelle homeostasis, remain uncertain. The present study demonstrates that in budding yeast, HU induces the formation of disulfide bonds in ER luminal misfolded proteins and specifically inhibits ERAD-L, an effect that is independent of S-phase arrest. In contrast, HU did not affect the ERAD-M and ERAD-C pathways or the degradation of cytosolic proteins. Moreover, HU did not affect the formation of the Hrd1 complex or the solubilities of ERAD-L substrates. Rather, the specific inhibition of ERAD-L is most likely due to the formation of disulfide bonds in the misfolded substrates, whose disulfide bonds must be reduced before retrotranslocation to the cytosol (Zhao et al, 2025). Furthermore, we discovered that HU mitigates the reductive stress phenotypes observed in cells lacking Ero1, which is essential for oxidative protein folding in the ER. These phenotypes include the ER-to-Golgi transport defects and cellular growth defects. Based on

these findings, we propose that HU acts as a modulator of thiol–disulfide homeostasis.

An intriguing finding of this study was that both HU and diamide were observed to specifically inhibit ERAD-L. In addition, along with those of previous studies (Frand & Kaiser, 1998), our results indicate that these drugs can mitigate the oxidative folding defect resulting from an Ero1 mutation by facilitating the formation of disulfide bonds between cysteine residues. The sole distinction between HU and diamide was that HU did not elicit UPR by itself, whereas diamide itself markedly induced UPR (Fig S14). Furthermore, the diamide-induced UPR has also been implicated in other literature (Han et al, 2010). We note that HU reportedly generates ROS possibly through one or more cellular metabolic intermediates, which is probably initiated by its hydroxylamine group (DeSesso, 1979; Huang et al, 2016; Shaw et al, 2024). Moreover, another study suggested that oxidative stress agents, such as $H_2O_2$ and trivalent arsenic (arsenite), can induce the cysteine-832–mediated inhibition of Ire1 in yeast (Guerra-Moreno et al, 2019). Consequently, one possible scenario is that ROS generated by HU treatment may also partially contribute to the inhibition of Ire1-mediated *HAC1* splicing.

In contrast, the ROS generated by HU treatment appears to be ineffective in inhibiting proteasome activity, as evidenced by the lack of impact on the degradation of ERAD-M, ERAD-C, and cytosolic substrates. This outcome contrasts with the impact of externally added $H_2O_2$, which impeded all ERAD pathways and the degradation of cytosolic substrates (see Fig S8A–F). It has been demonstrated that oxidative stress may exert a direct influence on the activities of the ubiquitin–proteasome system (Ishii et al, 2005; Gomes et al, 2006; Yang et al, 2007; Livnat-Levanon et al, 2014; Qiu et al, 2022). However, HU does not appear to lead to these phenomena because the degradation of ERAD-M, ERAD-C, and cytosolic substrate was unaffected by HU. It can thus be postulated that the ROS generated by HU treatment may differ in both quantity and quality from those produced by the externally added $H_2O_2$. Indeed, HU has been previously suggested to be degraded over time and in the presence of heat to generate a variety of chemicals, including *N*-hydroxyurethane, hydrogen cyanide, nitric oxide, and peroxides (Kuong & Kuzminov, 2009; Spivak & Hasselbalch, 2011; Fraser et al, 2015; Hallmark et al, 2021; Musialek & Rybaczek, 2021). It will therefore be important to analyze the nature of the ROS species generated by HU and to clarify which ROS species might contribute to the disulfide bond formation in the ER luminal proteins and to the specific inhibition of the ERAD-L.

A recent study (Sánchez-Molina et al, 2024 *Preprint*) indicates that HU induces thiol stress in the ER of the fission yeast *Schizosaccharomyces pombe*. The authors demonstrated that (1) the thiol-specific ER stress response triggers a transient morphological change in the ER and (2) ER expansion caused by thiol stress leads to the accumulation of the ER-specific Hsp70 chaperone Bip1, whereas several other heat shock proteins aggregate in cytoplasmic foci. They suggested that HU induces disulfide stress, which impairs proper protein folding in both the ER and cytoplasm. Although their study provides valuable insights into HU's effects in fission yeast and may complement our findings, our study provides direct evidence that HU modulates disulfide bond formation in ER luminal proteins. This modulation specifically inhibits the ERAD-L pathway under normal redox conditions while alleviating reductive

stress in cells deficient in Ero1. Overall, the biochemical and genetic approaches employed in our study contribute to a deeper understanding of HU's role in regulating organelle homeostasis.

HU was initially documented to possess antitumor properties in the 1960s and has since become a well-established, cost-effective therapeutic option for a spectrum of diseases, including brain tumors (Madaan et al, 2012), chronic myeloproliferative disorders (Spivak & Hasselbalch, 2011), and sickle cell anemia (SCA) (Platt et al, 1984; Tshilolo et al, 2019; Rankine-Mullings & Nevitt, 2022). In addition, HU has been demonstrated to diminish the adhesion of blood cells to vascular endothelial cells (Bartolucci et al, 2010; Chaar et al, 2014). Recently, HU has been demonstrated to enhance spatial memory in a mouse model of Alzheimer's disease, establishing it as a promising candidate for the treatment of cognitive decline associated with this disease (Brose et al, 2018). However, the precise mechanisms by which HU exerts its beneficial effects on these diseases remain unclear. Our results indicate that HU induces disulfide bond formation and alleviates oxidative folding defects in yeast ER. Future research from the perspective of thiol–disulfide homeostasis in the ER may prove beneficial in elucidating the mechanism of action of HU on various diseases.

# Materials and Methods

### Plasmids, strains, and culture conditions

Plasmids and yeast strains used in this study are listed in Tables S1 and S2, respectively. Yeast strains were grown in YPD+5xAde (1% yeast extract, 1% peptone, 2% glucose, and adenine hydrochloride [100 mg/liter]) or SD+DO+5xAde (0.67% yeast nitrogen base without amino acids, 2% glucose, dropout amino acids, and adenine hydrochloride [100 mg/liter]). Appropriate amino acids were removed from SD+DO+5xAde medium to maintain plasmids. Where indicated, 2% glucose was replaced with 2% galactose (Gal) or 2% raffinose (Raf). For agar plates, 2% agarose was included in the medium.

### Cycloheximide chase analysis

The cycloheximide chase protein degradation assay was performed as described previously (Nishio et al, 2023) with modifications. Where otherwise indicated, cells were inoculated into SD+DO+5-xAde, grown overnight, diluted into the same medium, and grown overnight to a log phase ($OD_{600}$ = 0.3 – 0.8). An equal number of cells were collected, resuspended into YPD+5xAde, and further grown for 1 h before hydroxyurea (#089-06651; FUJIFILM Wako), $H_2O_2$ (#086-07445; FUJIFILM Wako), diamide (#015-25353; FUJIFILM Wako), *N*-acetylcysteine (NAC) (#015-05132; FUJIFILM Wako), or MMS (#M-0369; Tokyo Chemical Industry) was added to the medium under the conditions described in the figure legends. When the cell cycle was synchronized, cells were grown to a log phase in SD+DO+5xAde and washed twice with distilled water. The medium was changed to YPD+5xAde, and the cell density was adjusted to $OD_{600}$ = 0.3 before they were incubated with 10 $\mu$g/ml $\alpha$-factor (chemically synthesized by GenScript), 10 mg/ml hydroxyurea, or 15 $\mu$g/ml nocodazole (#140-08531; FUJIFILM Wako) for 2.5 h.

Cycloheximide (#06741-04; Nacalai Tesque) was added to the medium at a final concentration of 200 $\mu$g/ml to terminate protein synthesis. At the indicated time points, cells were collected and mixed with 10% TCA on ice. Cells were then collected by centrifugation, and the pellet was stored at –80°C. Subsequently, the cell pellet was suspended in 300 $\mu$l of 20% TCA and lysed by vigorous vortexing with glass beads for ~30 min (Max Mixer EVR-032; TAITEC), with occasional inversion of the tube to prevent the accumulation of cells at the bottom. The broken cell lysate was added to 900 $\mu$l of 5% TCA, the tube was inverted several times, and 1 ml of the suspension was transferred to a new tube. Proteins were precipitated by centrifugation at 20,000$g$ for 10 min at 4°C. The protein pellets were dissolved in KNTCASB (80 mM Tris–HCl [pH 7.5], 8 mM EDTA [pH 8.0], 12.5% glycerol, 8 M urea, 4% SDS, 200 mM DTT, Tris [0.8 mg/ml], and 0.1% bromophenol blue [BPB]). All samples were heated at 55°C for 15 min and cleared by centrifugation at 20,000$g$ for 1 min at RT before SDS–polyacrylamide gel electrophoresis (SDS–PAGE) and Western blotting.

### Co-immunoprecipitation assay

Cells were grown to a mid-log phase ($OD_{600}$~1.5). Cells (~100 $OD_{600}$) were harvested and disrupted with glass beads in lysis buffer (20 mM HEPES-KOH, pH 7.1, 50 mM KOAc, 2 mM EDTA, and 100 mM sorbitol) supplemented with cOmplete, EDTA-free Protease Inhibitor Cocktail (#COEDTAF-RO; Merck) (PIC) in a round-bottom plastic tube by eight rounds of vigorous vortexing for 30 s with 30 s interval on ice. The cell lysate was collected by adding 400 $\mu$l of buffer 88 (20 mM Hepes, pH 7.1, 150 mM KOAc, 250 mM sorbitol, 5 mM MgOAc) with PIC and transferred to a microfuge tube. The glass beads were washed with 300 $\mu$l of buffer 88 with PIC, and the lysate was pooled in the same tube. After unbroken cells were removed by centrifugation at 300$g$ for 5 min at 4°C, the supernatant was further centrifuged at 20,000$g$ for 20 min at 4°C. The pellet (membrane) fractions were solubilized in a buffer (20 mM Tris–HCl, pH 7.5, 100 mM NaCl, and 10% glycerol) supplemented with either 1% digitonin (#047-21374; FUJIFILM Wako) (Hrd1-3FLAG) or 0.5% Triton X-100 (CPY*-3HA). The solubilized membranes were cleared by centrifugation at 20,000$g$ for 20 min at 4°C. Proteins were immunoprecipitated with anti-FLAG M2 affinity gel (#A2220; Sigma-Aldrich) or anti-HA antibody (#180-3 TANA2; MBL) bound to Dynabeads Protein G (Invitrogen). After nutation at 4°C for 1–2 h, the gel or beads were washed three times with solubilization buffer supplemented with 0.5% digitonin or 0.5% Triton X-100 before proteins were eluted with KNTCASB.

### Aggregation assay

Cells were inoculated into SD+DO+5xAde and grown overnight at 26°C. Cells were diluted into 50 ml of the same medium and grown overnight at 26°C until $OD_{600}$ reached ~0.5. HU was added to the medium at the final concentration of 10 mg/ml, and cells were further grown at 37°C for 2 h. Cells (40 $OD_{600}$ equivalent) were collected and washed with distilled water before they were lysed in lysis buffer (20 mM HEPES-KOH, pH 7.1, 150 mM KOAc, 2 mM EDTA, 100 mM sorbitol) supplemented with PIC using glass beads in a round-bottom plastic tube by eight rounds of vigorous vortexing for

30 s with 30 s interval on ice. The cell lysate was collected by adding 400 μl of buffer 88 with PIC and transferred to a microfuge tube. The glass beads were washed with 300 μl of buffer 88 with PIC, and the lysate was pooled in the same tube. The collected lysate was cleared by centrifugation at 300*g* for 5 min at 4°C to remove cell debris and unbroken cells before 700 μl of supernatant was transferred to a low-binding microfuge tube (#PK-15C-500; WAT-SON). The cleared lysate was further centrifuged at 20,000*g* for 20 min at 4°C. The pellet fraction containing microsomes was detergent-solubilized in a buffer (20 mM Tris–HCl, pH 7.5, 150 mM NaCl, 1% Triton X-100, 10 mM EDTA) for 30 min on ice and further centrifuged at 20,000*g* for 10 min at 4°C. The supernatant (800 μl) was subjected to the ultracentrifugation in a TLA-100.3 rotor (Beckman Coulter, Optima MAX-XP) at 100,000*g* for 30 min at 4°C (P100 fraction). Proteins in the supernatant (S100 fraction) were precipitated with 16% TCA and rinsed with ice-cold acetone before being solubilized in KNTCASB. The P100 pellet was solubilized in KNTCASB. Both samples were heated at 55°C for 15 min and were subjected to Western blotting analysis.

### Modification of Cys with maleimide-PEG5000

The maleimide-PEG5000 modification assay was performed as described previously (Tsai & Rapoport, 2002; Sakoh-Nakatogawa et al, 2009) with modifications. Cells were grown in SD+DO+5xAde overnight until $OD_{600}$ reached 0.5–0.6. Equal numbers of cells were collected, resuspended in YPD+5xAde, and further grown for 1 h before further incubation with chemicals for the indicated time periods as described in the figure legends. The culture medium containing 3–5 $OD_{600}$ equivalent of cells was directly mixed with TCA at the final concentration of 20%. Cells were collected by centrifugation and washed with 20% TCA before they were suspended in 300 μl of 20% TCA and stored at −80°C. Cells were disrupted with glass beads, and the protein pellets were prepared as described in the section "Cycloheximide chase analysis." Protein pellets were further washed twice with ice-cold acetone and suspended in nonreducing SDS sample buffer (80 mM Tris–HCl, pH 6.8, 4% SDS, 8 M urea, 12.5% glycerol, trace amount of BPB) supplemented with 5 mM maleimide-PEG5000 (#MF001022-5K; Biopharma PEG) before the reaction mixture was incubated for 30 min at 37°C. We note that nonreducing SDS sample buffer should be made fresh every time. Reduced and oxidized controls were prepared by incubating the protein pellets in nonreducing SDS sample buffer supplemented with either 100 mM DTT or 5 mM 5,5′-dithiobis(2-nitrobenzoic acid) (#043-16403; FUJIFILM Wako) for 30 min at 37°C. In this case, proteins were precipitated again with 20% TCA, washed with ice-cold acetone twice, and resuspended in nonreducing SDS sample buffer supplemented with maleimide-PEG5000. The modification reaction was quenched by adding 50 mM DTT. Samples were then incubated at 55°C for 15 min before they were subjected to SDS–PAGE and Western blotting analysis.

### Immunoblotting

Proteins were transferred from polyacrylamide gels to Immobilon-P membranes (Millipore) in blotting buffer (25 mM Tris, 192 mM glycine, and 10% methanol) using the GENIE electrophoretic transfer device (Idea Scientific Company) at a constant current of 650 mA. The membranes were washed with TBS-T buffer (20 mM Tris–HCl, pH 7.5, 150 mM NaCl, and 0.1% Tween-20) and, when necessary, blocked with 3% skimmed milk in TBS-T buffer for 30 min. Subsequently, the membranes were incubated with primary anti-bodies in either TBS-T buffer or TBS-T buffer supplemented with 3% skimmed milk at 4°C overnight. The membranes were then washed three times with TBS-T (10–60 min per wash), incubated with secondary antibodies for ~60 min, and washed three times with TBS-T. Last, the membranes were incubated with Chemi-Lumi One L (#07880; Nacalai Tesque) or Luminata Forte Western HRP Substrate (# WBLUF0500; MERCK) and exposed to x-ray film. Band intensities were quantified with ImageJ (National Institutes of Health [NIH]) directly from films scanned at high resolution (#GT-X980; EPSON) in TIF file format (600 dpi). In some figures, the chemiluminescence signals were detected by FUSION SOLO S (VILBER) and quantified with ImageJ.

Anti-HA antibody was purchased from MBL (#180-3; TANA2). Anti-Sic1, anti-Clb2, and anti-myc antibodies were generous gifts from T. Kamura (Nagoya University). Anti-Pgk1 antibody was purchased from Abcam (#ab113687, 22C5D8). Anti-Cdc48, anti-Pdi1, and anti-CPY antibodies were generous gifts from T. Endo (Kyoto Sangyo University). Anti-Pdi1 antibody purchased from Abcam (#ab4644, 38H8) was also used. Anti-Hrd3p antibody was a generous gift from T. Sommer and J. Ernst (Max-Delbrück-Center for Molecular Medi-cine). Anti-Kar2p antibody was a generous gift from R. Schekman (U.C. Berkeley). Anti-Spo12 antibody was described previously (Nakatsukasa et al, 2018). Anti-Hrd1, anti-Usa1, anti-Der1, and anti-Yos9 antibodies were described previously (Nakatsukasa et al, 2013). Anti-Rad53 antibody was purchased from Abcam (#ab104232).

### UPR assay

The UPR assay was performed essentially as described previously (Burke et al, 2000) with minor modifications. Cells carrying pJC104 (encoding *UPRE-LacZ* reporter) (Cox & Walter, 1996) were grown to $OD_{600}$ = 0.6 – 1.2 in SD+DO+5xAde. Where indicated, hydroxyurea or DTT was added to the culture as described in figure legends. One mL of culture was added to 400 μl of ice-cold YPD+5xAde to quantitatively collect cells. Cells were then collected by cen-trifugation at 20,000*g* for 1 min at 4°C and stored at −80°C. The cell pellets were suspended in 800 μl of β-galactosidase reaction buffer (100 mM sodium phosphate buffer, pH 7.0, 10 mM KCl, 1 mM $MgSO_4$, and 40 mM 2-mercaptoethanol). The suspension was added to 20 μl of 0.1% SDS and 50 μl of chloroform before cells were lysed by vigorous vortexing for 1 min and incubated at 28°C. The cell lysates were mixed with 200 μl of 4 mg/ml 2-nitrophenyl-β-D-galactopyranoside (ONPG, the β-galactosidase substrate) solution. The β-galactosidase reaction (conversion of ONPG to 2-nitro-phenol) was performed at 28°C. The reaction was terminated by adding 500 μl of 1 M $Na_2CO_3$ to the reaction mixture. After being clarified by centrifugation at 8,000*g* for 5 min, optical absorbance at 420 nm ($A_{420}$) and 550 nm ($A_{550}$) was measured to monitor con-centration of the reaction product (2-nitrophenol). The cellular relative β-galactosidase activity was calculated using the formula: ($A_{420}$ after the reaction-1.75 × $A_{550}$ after the reaction)/[($OD_{600}$ of the culture) × (reaction time)].

The *HAC1* splicing was measured as described previously (Tran et al, 2019) with minor modifications. Cells were grown to $OD_{600}$ = 0.5 in SD+DO+5xAde before they were incubated with chemicals, and/or the temperature was shifted as described in figure legends. Cells were then collected by centrifugation and snap-frozen in liquid nitrogen before they were stored at −80°C. Total RNA was isolated with RNeasy Mini Kit (#74104; QIAGEN). Genomic DNA was digested with RNase-Free DNase Set (#79254; QIAGEN) during the procedure. The RNA samples were then used for reverse transcriptase (RT)–PCR amplification of the *HAC1* transcripts using a polyA RT primer and *HAC1*-specific PCR primers (Tran et al, 2019). The PCR products were then analyzed by 2% agarose gel electrophoresis (TBE buffer), and ethidium bromide fluorescent images of the resulting gels were pictured with a Gel Doc XR+ imaging system (Bio-Rad). The band intensities were quantified to calculate the values of the *HAC1* mRNA-splicing efficiency using the formula: 100 × (band intensity of *HAC1*i)/ [(band intensity of *HAC1*i) + (band intensity of *HAC1*u)], where *HAC1*i and *HAC1*u respectively are the spliced and unspliced forms of the *HAC1* mRNA.

## Data Availability

No datasets were generated during the current study. All data are available in the main text or the supporting information.

## Supplementary Information

## Acknowledgements

We thank the National BioResource Project for plasmids; and Jeffrey L Brodsky, Toshiya Endo, Randolph Y Hampton, Ernst Jarosch, Takumi Kamura, Chris Kaiser, Jaekwon Lee, Susan Michaelis, Davis TW Ng, Randy Schekman, Thomas Sommer, Peter Walter, Jonathan S Weissman, and Jakob R Winther for plasmids, strains, and antibodies. We acknowledge the assistance of the Research Equipment Sharing Center at Nagoya City University. We also thank the Nakatsukasa laboratory members for discussions and technical assistance. This work was supported by grants from Japan Society for the Promotion of Science KAKENHI 19H02923, 23K26869, and 24H01906, Hori Sciences and Arts Foundation, and Institute for Fermentation, Osaka, to K Nakatsukasa; and Japan Science and Technology Agency SPRING grant JPMJSP2130 to Y Takano.

### Author Contributions

Y Takano: resources, formal analysis, funding acquisition, investigation, methodology, and writing—original draft, review, and editing.
Y Ishiwata-Kimata: resources, investigation, and methodology.
R Ushioda: conceptualization.
Y Kimata: resources, investigation, and methodology.
K Nakatsukasa: conceptualization, resources, formal analysis, supervision, funding acquisition, investigation, methodology, project administration, and writing—original draft, review, and editing.

### Conflict of Interest Statement

The authors declare that they have no conflict of interest.

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
