## [Reviewer comments · Life Science Alliance]

Life Science Alliance

Hydroxyurea modulates thiol-disulfide homeostasis in the yeast endoplasmic reticulum

Yuki Takano, Yuki Ishikawa-Kimata, Ryo Ushioda, Yukio Kimata, and Kunio Nakatsukasa

DOI: <https://doi.org/10.26508/lsa.202503225>

Corresponding author(s): Kunio Nakatsukasa, Nagoya City University

Review Timeline:

Submission Date:	2025-01-14
Editorial Decision:	2025-02-28
Revision Received:	2025-04-28
Editorial Decision:	2025-05-28
Revision Received:	2025-05-29
Accepted:	2025-05-29

Scientific Editor: Tim Fessenden

Transaction Report:

February 28, 2025

Re: Life Science Alliance manuscript #LSA-2025-03225-T

Prof. Kunio Nakatsukasa
Nagoya City University
Japan

Dear Dr. Nakatsukasa,

Thank you for submitting your manuscript entitled "Hydroxyurea modulates thiol-disulfide homeostasis in the yeast endoplasmic reticulum" to Life Science Alliance. The manuscript was assessed by expert reviewers, whose comments are appended to this letter. We invite you to submit a revised manuscript addressing the Reviewer comments.

Thank you for this interesting contribution to Life Science Alliance. We are looking forward to receiving your revised manuscript.

Sincerely,

B. MANUSCRIPT ORGANIZATION AND FORMATTING:

Reviewer #1 (Comments to the Authors (Required)):

In their manuscript „Hydroxyurea modulates thiol-disulfide ..." Takano and colleagues report on the oxidation of secretory proteins by the therapeutically important drug hydroxyurea (HU) in yeast. They start with the observation that HU inhibits the turnover of soluble misfolded proteins from the lumen of the endoplasmic reticulum (ER), which are destined for proteolysis by a cellular mechanism termed ER-associated degradation (ERAD). Intriguingly, the processing of membrane-bound ERAD client proteins is not affected by HU. Interference of ERAD by HU is independent from the documented activity of HU to arrest cells in the S-phase of the cell cycle. In a biochemical assay, the authors then observe that HU treatment substantially supports the formation of disulfide bonds in misfolded secretory proteins. Importantly, HU alleviates the protein transport defects observed in yeast cells lacking Ero1 function, a protein disulfide oxidase that promotes disulfide bond formation in the ER lumen by oxidizing the protein disulfide isomerase Pdi1. Indeed, when compared to untreated yeast, administering HU on Ero1-defective cells give only a mild induction of the UPR, a cellular stress pathway that senses misfolded proteins in the ER lumen. These finding and additional biochemical experiments strongly imply that HU fosters oxidative folding of secretory proteins in the ER lumen and can partly compensate the loss of Ero1. The degradation of misfolded ERAD substrate is inhibited by HU because disulfide bridges in these proteins have to be reduced to enable transport into the cytoplasm for further processing. Concordantly, ERAD of a soluble client protein lacking internal cysteine residues is much less affected by HU than that of a variant containing disulfide bridges.

The authors present a wealth of data that quite convincingly support their point of view. All experiments are well done and include the necessary controls. The text is clear, but the discussion is a bit long and should be shortened to improve readability. The novel finding of HU as a compound that promotes oxidative protein biogenesis in the ER has potential implications for the treatment of patients. I have only a minor question that should be addressed to allow publication in "Life Science Alliance".

The authors have to use rather high dosages of HU (5 - 10 mM) to observe significant effects on the degradation of the ERAD substrate and on the biogenesis of secretory proteins in yeast cells lacking Ero1 activity. This may either be due to a rather unspecific activity of HU in oxidative protein folding or to the export of HU via transport proteins such as the ABC transporter Pdr5. It would thus be of interest whether in absence of the multidrug resistance transporter Pdr5 lower amounts of HU suffice to e.g. restore growth of ero1-1 cells at non-permissive temperature.

The criticism of Reviewer 2 is justified, but in my opinion somewhat exaggerated. The authors have sufficiently shown that the cell cycle defect caused by HU is not the main cause of the observed degradation defect of ER luminal proteins. Their alternative explanation of stabilization of disulfide bridges in secretory proteins by HU is, in my opinion, quite well supported by their results. The experiments suggested by Reviewer 2 would of course provide additional support for the authors' claims, but are not absolutely necessary in my opinion.

Reviewer #2 (Comments to the Authors (Required)):

Hydroxyurea (HU) is used to induce S-phase arrest and checkpoint activation. The authors examined the effect of HU treatment of yeast cells on organelle homeostasis and found that HU specifically inhibits ERAD-L, independently of S-phase arrest, but has no effect on ERAD-M, ERAD-C, or degradation of cytosolic proteins, which is quite interesting.

Major critiques

- 1) Independence from S-phase arrest; the authors claimed that CPY* was degraded in S-phase (Fig. 2C, lower panel, lanes 9-12), however, after 40 min chase the cells were considered to be in G1 from the data shown in Fig. 2B (30 min + 40 min = 70 min), making the authors' claim inappropriate. Rather, the results of MMS treatment (Fig. S4D) would become more important if the authors could show that MMS treatment actually activates the S-phase checkpoint (these data should be shown in Fig. 2).
- 2) Degradation of CPY*CysLess; CPY*CysLess was weakly stabilized in HU-treated cells, in contrary to the authors' expectation (Fig. S13A). The authors consider that endogenous, oxidized ERAD substrates accumulates in the ER lumen of HU-treated cells, which block retrotranslocation of CPY*CysLess. How about the effect of diamide on the degradation of CPY*CysLess? Is it similar to the effect of HU? What happens if CPY*CysLess-expressing cells were treated with H₂O₂? Does the degradation of CPY*CysLess become slower? It is because H₂O₂ inhibits not only ERAD-L but also ERAD-M, ERAD-C, and degradation of cytosolic proteins (Fig. S8).
- 3) Why can WT and ero1-1 cells grow in the presence of 10 mM and 50 mM HU (Fig. 4A)? It should block cell cycle progression.

4) UPR; What happens to HAC1 mRNA splicing and UPR-reporter expression if *ero1-1* cells are treated with diamide?

Minor points

- 1) Fig. S2; describe in the legend what is the upper band of Clb2 observed in cells treated with NC (lanes 4, 8, 12 in A; lanes 4, 8, 12 and 16 in B).
- 2) Fig. S4; describe in the legend what are tom1 cdh1 .
- 3) Fig. S5B; add *hrd1* at the top of the panel and add explanation that *hrd1* cells were transformed by Hrd1-3FLAG or untagged Hrd1 to the legend.
- 4) Fig. S5C; remove Yos9 from the panel or describe it as endogenous Yos9.
- 5) Fig. S5D; remove Hrd3 from the panel or describe it as endogenous Hrd3.

Point-by-point responses to the reviewers' comments

We appreciate the reviewer's excellent comments. As detailed below, we have addressed each of the Major and Minor points.

Reviewer #1:

In their manuscript "Hydroxyurea modulates thiol-disulfide ..." Takano and colleagues report on the oxidation of secretory proteins by the therapeutically important drug hydroxyurea (HU) in yeast. They start with the observation that HU inhibits the turnover of soluble misfolded proteins from the lumen of the endoplasmic reticulum (ER), which are destined for proteolysis by a cellular mechanism termed ER-associated degradation (ERAD). Intriguingly, the processing of membrane-bound ERAD client proteins is not affected by HU. Interference of ERAD by HU is independent from the documented activity of HU to arrest cells in the S-phase of the cell cycle. In a biochemical assay, the authors then observe that HU treatment substantially supports the formation of disulfide bonds in misfolded secretory proteins. Importantly, HU alleviates the protein transport defects observed in yeast cells lacking Ero1 function, a protein disulfide oxidase that promotes disulfide bond formation in the ER lumen by oxidizing the protein disulfide isomerase Pdi1. Indeed, when compared to untreated yeast, administering HU on Ero1-defective cells give only a mild induction of the UPR, a cellular stress pathway that senses misfolded proteins in the ER lumen. These finding and additional biochemical experiments strongly imply that HU fosters oxidative folding of secretory proteins in the ER lumen and can partly compensate the loss of Ero1. The degradation of misfolded ERAD substrate is inhibited by HU because disulfide bridges in these proteins have to be reduced to enable transport into the cytoplasm for further processing. Concordantly, ERAD of a soluble client protein lacking internal cysteine residues is much less affected by HU than that of a variant containing disulfide bridges.

The authors present a wealth of data that quite convincingly support their point of view. All experiments are well done and include the necessary controls. The text is clear, but the discussion is a bit long and should be shortened to improve readability. The novel finding of HU as a compound that promotes oxidative protein biogenesis in the ER has potential implications for the treatment of patients. I have only a minor question that should be addressed to allow publication in "Life Science Alliance".

We appreciate the reviewer's positive comments on our manuscript. The discussion section has been slimed to improve readability.

The authors have to use rather high dosages of HU (5 - 10 mM) to observe significant effects on the degradation of the ERAD substrate and on the biogenesis of secretory proteins in yeast cells lacking Ero1 activity. This may either be due to a rather unspecific activity of HU in oxidative protein folding or to the export of HU via transport proteins such as the ABC transporter Pdr5. It would thus be of interest whether in absence of the multidrug resistance transporter Pdr5 lower amounts of HU suffice to e.g. restore growth of ero1-1 cells at non-permissive temperature.

We thank the reviewer for this valid suggestion. We investigated the varying degrees of HU concentration capable of rescuing the temperature-sensitive growth defect observed for *ero1-1* cells and found that 1 mM HU was able to rescue the growth defect (Fig. S14A). We then constructed *ero1-1pdr5Δ* strain and analyzed its growth on the medium containing HU. In the absence of Pdr5, a multidrug transporter, 1 mM HU exhibited a slightly more pronounced effect on rescuing the growth defect (Fig. S14A). This phenomenon can be attributed to the absence of exportation of HU from the cells in the absence of Pdr5, resulting in its accumulation within the cell. Interestingly, at high concentrations of HU (5~10 mM), the growth levels of the *ero1-1pdr5Δ* cells were found to be lower than that of *ero1-1* cells (Fig. S14A). One potential explanation for this phenomenon is that the high concentration of HU caused delayed S-phase progression in cells lacking Pdr5 with greater efficiency compared to cells harboring Pdr5. However, *ero1-1* and *ero1-1pdr5Δ* cells exhibited comparable growth patterns even in the presence of 5~10 mM HU at a permissive temperature of 26°C, implying that there is not a considerable difference in the delayed S-phase progression between these strains. Regardless of the precise reason for this phenomenon, it is posited that the deletion of a drug transporter such as Pdr5 may exert a more potent effect on the efficacy of HU as an oxidant for the ER luminal proteins.

The criticism of Reviewer 2 is justified, but in my opinion somewhat exaggerated. The authors have sufficiently shown that the cell cycle defect caused by HU is not the main cause of the observed degradation defect of ER luminal proteins. Their alternative explanation of stabilization of disulfide bridges in secretory proteins by HU is, in my opinion, quite well supported by their results. The experiments suggested by Reviewer 2 would of course provide additional support for the authors' claims, but are not absolutely necessary in my opinion.

Thank you for the positive comments on our study.

Reviewer #2:

Hydroxyurea (HU) is used to induce S-phase arrest and checkpoint activation. The authors examined the effect of HU treatment of yeast cells on organelle homeostasis and found that HU specifically inhibits ERAD-L, independently of S-phase arrest, but has no effect on ERAD-M, ERAD-C, or degradation of cytosolic proteins, which is quite interesting.

Major critiques

1) Independence from S-phase arrest; the authors claimed that CPY* was degraded in S-phase (Fig. 2C, lower panel, lanes 9-12), however, after 40 min chase the cells were considered to be in G1 from the data shown in Fig. 2B (30 min + 40 min = 70 min), making the authors' claim inappropriate. Rather, the results of MMS treatment (Fig. S4D) would become more important if the authors could show that MMS treatment actually activates the S-phase checkpoint (these data should be shown in Fig. 2).

We thank the reviewer for this valid suggestion. First, we concur with the reviewers' assertion that cells may be regarded as being in G1 after a 40-minute chase. However, it is noteworthy that the initial rate of CPY* degradation in these cells was faster than that in cells treated with HU (Fig. 2C). Consequently, this data may well corroborate our assertion that CPY* degradation is independent of S-phase arrest. In the main text, the following sentence was added: "In these cells, the initial rate of CPY* degradation was faster than that in cells treated with HU."

Second, when cells were treated with MMS or HU under the conditions of Fig. 2D, we did indeed observe phosphorylation of Rad53 (Fig. 2E), indicating that MMS treatment in Fig. 2D actually activated the S-phase checkpoint. However, CPY* degradation was unaffected in cells treated with MMS, whereas it was considerably inhibited in HU-treated cells (Fig. 2D). Therefore, we believe that the S-phase checkpoint activation by HU is not a primary reason for the specific inhibition of ERAD-L.

2) Degradation of CPY*CysLess; CPY*CysLess was weakly stabilized in HU-treated cells, in contrary to the authors' expectation (Fig. S13A). The authors consider that endogenous, oxidized ERAD substrates accumulates in the ER lumen of HU-treated cells, which block retrotranslocation of CPY*CysLess. How about the effect of diamide on the degradation of CPY*CysLess? Is it similar to the effect of HU? What happens if CPY*CysLess-expressing cells were treated with H₂O₂? Does the degradation of CPY*CysLess become slower? It is because H₂O₂ inhibits not only ERAD-L but also ERAD-M, ERAD-C, and degradation of cytosolic proteins (Fig. S8).

We thank the reviewer for this valid suggestion. As shown in Fig. S13A, CPY*CysLess was weakly stabilized in diamide-treated cells to the same extent as in cells treated with HU. We also note that CPY*CysLess degradation was considerably slowed in cells treated with H₂O₂ (Fig. S13A). This is likely because H₂O₂ inhibits the degradation of all pathways tested in this study; not only ERAD-L but also ERAD-M, ERAD-C, and degradation of cytosolic proteins.

3) Why can WT and *ero1-1* cells grow in the presence of 10 mM and 50 mM HU (Fig. 4A)? It should block cell cycle progression.

In order to achieve synchronization of the yeast cell cycle at the S-phase, cells are typically exposed to a relatively high concentration of hydroxyurea (HU) (10 mg/ml = 132 μM). It is my understanding that HU delays the cell cycle and retards cell growth; however, its effect is not complete. Indeed, in numerous published literatures, the plate-based yeast growth assay (serial dilution assay), such as the one performed in Fig. 4A in the current study, has been performed in the presence of even 100 μM HU. In this case, cell growth is somewhat slowed. For further elucidation, please refer to one of the following examples in recent literature:

<https://www.life-science-alliance.org/content/5/4/e202101153>

Consequently, we posit that the utilization of 10-50 μM HU in the serial dilution assay is a

rational approach.

4) UPR; What happens to HAC1 mRNA splicing and UPR-reporter expression if *ero1-1* cells are treated with diamide?

We thank the reviewer for this valid suggestion. At concentrations of diamide as low as 27.8 μM and 166.7 μM , diamide was observed to suppress the UPR of *ero1-1* cells. Intriguingly, at the higher concentration (1000 μM), it caused the UPR of *ero1-1* cells (Fig. S14G). In case of diamide, there may be a threshold of approximately 200 μM exists. At concentrations lower than the threshold, the oxidative capacity of diamide may function to rescue the reductive stress of *ero1-1* cells. However, at higher concentrations, excess diamide may trigger oxidation of the ER luminal proteins in *ero1-1* cells, consequently activating the UPR.

Minor points

1) Fig. S2; describe in the legend what is the upper band of Clb2 observed in cells treated with NC (lanes 4, 8, 12 in A; lanes 4, 8, 12 and 16 in B).

The upper band of Clb2 observed in cells treated with NC is likely a phosphorylated form of Clb2. As indicated by the following publication, such posttranslational modification has been observed in other studies:

<https://pmc.ncbi.nlm.nih.gov/articles/PMC3466553/>

This has been mentioned in the figure legend in Fig. S2.

2) Fig. S4; describe in the legend what are *tom1* and *cdh1*;

Spo12 degradation is reportedly inhibited in cells lacking Tom1 ubiquitin ligase during G2/M phase. Its degradation is also inhibited in cells lacking Cdh1, an activator of the anaphase-promoting complex, during G1 phase (Nakatsukasa et al., FEBS letters 2018).

<https://febs.onlinelibrary.wiley.com/doi/full/10.1002/1873-3468.13066>

Thus, the double mutant has been used as a control. These have been mentioned in the figure legend.

3) Fig. S5B; add *hrd1*; at the top of the panel and add explanation that *hrd1*; cells were transformed by Hrd1-3FLAG or untagged Hrd1 to the legend.

In lanes 1 and 4 in Fig. S5B, wild-type cells were used. In lanes 2, 3, 5, and 6, 3xFLAG tag was integrated at the C-terminus of the endogenous locus of Hrd1. Hrd1-3FLAG was expressed from its endogenous locus in the chromosome. Therefore, the term of "endogenous" has been incorporated in the figure.

4) Fig. S5C; remove Yos9 from the panel or describe it as endogenous Yos9.

As suggested, Yos9 has been removed from the panel.

5) Fig. S5D; remove Hrd3 from the panel or describe it as endogenous Hrd3.

As suggested, Hrd3 has been removed from the panel.

May 28, 2025

RE: Life Science Alliance Manuscript #LSA-2025-03225-TR

Prof. Kunio Nakatsukasa
Nagoya City University
Graduate School of Science
Yamanohata 1, Mizuho-cho, Mizuho-ku
Nagoya, Aichi 467-8501
Japan

Dear Dr. Nakatsukasa,

Thank you for submitting your revised manuscript entitled "Hydroxyurea modulates thiol-disulfide homeostasis in the yeast endoplasmic reticulum". As you will see below, the reviewers are satisfied with the revisions in place. We would be happy to publish your paper in Life Science Alliance pending final revisions necessary to meet our formatting guidelines.

- Please upload your main and supplementary figures as single files.
- Please add ORCID ID for the corresponding author - you should have received instructions on how to do so.
- Please add the X and Bluesky handles of your host institute/organization, as well as your own and/or one of the authors, in our system.
- Please add callouts for Figures S2A-B; S8A-F; S9A-F and S10A-F to your main manuscript text.
- There is a call-out for figure S15A. Please correct to S15 as this figure has only one panel.
- Please add molecular weight markers to the blots in Figure S13B.

A. FINAL FILES:

B. MANUSCRIPT ORGANIZATION AND FORMATTING:

per figure for this information. These files will be linked as supplementary "Source Data" files.

Sincerely,

Reviewer #1 (Comments to the Authors (Required)):

I was already rather positive on the first version of the work and the authors successfully addressed all my minor questions. I therefore support publication of the revised manuscript "Hydroxyurea modulates thiol-disulfide ..." in "Life Science Alliance".

Reviewer #2 (Comments to the Authors (Required)):

As the authors addressed my points adequately, I fully recommend its publication in Life Science Alliance.

Point-by-point responses to the reviewers' comments

Reviewer #1:

I was already rather positive on the first version of the work and the authors successfully addressed all my minor questions. I therefore support publication of the revised manuscript "Hydroxyurea modulates thiol-disulfide ..." in "Life Science Alliance".

N/A

Reviewer #2:

As the authors addressed my points adequately, I fully recommend its publication in Life Science Alliance.

N/A

May 29, 2025

RE: Life Science Alliance Manuscript #LSA-2025-03225-TRR

Prof. Kunio Nakatsukasa
Nagoya City University
Graduate School of Science
Yamanohata 1, Mizuho-cho, Mizuho-ku
Nagoya, Aichi 467-8501
Japan

Dear Dr. Nakatsukasa,

Thank you for submitting your Research Article entitled "Hydroxyurea modulates thiol-disulfide homeostasis in the yeast endoplasmic reticulum". It is a pleasure to let you know that your manuscript is now accepted for publication in Life Science Alliance. Congratulations on this interesting work.

DISTRIBUTION OF MATERIALS:

Again, congratulations on a very nice paper. I hope you found the review process to be constructive and are pleased with how the manuscript was handled editorially. We look forward to future exciting submissions from your lab.

Sincerely,
